# Homocystamide Conjugates of Human Serum Albumin as a Platform to Prepare Bimodal Multidrug Delivery Systems for Boron Neutron Capture Therapy

**DOI:** 10.3390/molecules26216537

**Published:** 2021-10-29

**Authors:** Tatyana Popova, Maya A. Dymova, Ludmila S. Koroleva, Olga D. Zakharova, Vladimir A. Lisitskiy, Valeria I. Raskolupova, Tatiana Sycheva, Sergei Taskaev, Vladimir N. Silnikov, Tatyana S. Godovikova

**Affiliations:** 1Institute of Chemical Biology and Fundamental Medicine, SB RAS, 630090 Novosibirsk, Russia; io197724@gmail.com (T.P.); maya.a.rot@gmail.com (M.A.D.); koroleva@niboch.nsc.ru (L.S.K.); garonna3@mail.ru (O.D.Z.); foxvo.vl@gmail.com (V.A.L.); v.raskolupova@mail.ru (V.I.R.); silnik@niboch.nsc.ru (V.N.S.); 2Faculty of Natural Sciences, Novosibirsk State University, 630090 Novosibirsk, Russia; taskaev@inp.nsk.su; 3Budker Institute of Nuclear Physics, SB RAS, 630090 Novosibirsk, Russia; sychevatatyanav@gmail.com

**Keywords:** boron neutron capture therapy, boron delivery agents, thenoyltrifluoroacetone, boronated albumin theranostic, conjugate, in vitro efficacy evaluation, irradiated by epithermal neutron flux, colony forming assay

## Abstract

Boron neutron capture therapy is a unique form of adjuvant cancer therapy for various malignancies including malignant gliomas. The conjugation of boron compounds and human serum albumin (HSA)—a carrier protein with a long plasma half-life—is expected to extend systemic circulation of the boron compounds and increase their accumulation in human glioma cells. We report on the synthesis of fluorophore-labeled homocystamide conjugates of human serum albumin and their use in thiol-‘click’ chemistry to prepare novel multimodal boronated albumin-based theranostic agents, which could be accumulated in tumor cells. The novelty of this work involves the development of the synthesis methodology of albumin conjugates for the imaging-guided boron neutron capture therapy combination. Herein, we suggest using thenoyltrifluoroacetone as a part of an anticancer theranostic construct: approximately 5.4 molecules of thenoyltrifluoroacetone were bound to each albumin. Along with its beneficial properties as a chemotherapeutic agent, thenoyltrifluoroacetone is a promising magnetic resonance imaging agent. The conjugation of bimodal HSA with undecahydro-*closo*-dodecaborate only slightly reduced human glioma cell line viability in the absence of irradiation (~30 μM of boronated albumin) but allowed for neutron capture and decreased tumor cell survival under epithermal neutron flux. The simultaneous presence of undecahydro-*closo*-dodecaborate and labeled amino acid residues (fluorophore dye and fluorine atoms) in the obtained HSA conjugate makes it a promising candidate for the combination imaging-guided boron neutron capture therapy.

## 1. Introduction

At present, cancer therapy includes surgery, chemotherapy, radiotherapy, targeted therapy, and immunotherapy. A newly emerging treatment option—boron neutron capture therapy (BNCT), in which drugs containing enriched boron are accumulated in tumor cells followed by their neutron beam radiation—offers an advantage over conventional chemo- and radiotherapies as it selectively targets tumor cells without significantly affecting healthy tissues. This is especially beneficial for clinical cases characterized by the infiltration of tumors into normal tissue or broad expansion of the tumor into the whole organ [1,2]. Clinical interest in BNCT has focused primarily on high-grade gliomas [3,4,5,6], patients with recurrent tumors of the head and neck region [7,8,9,10,11,12,13,14], which have failed conventional therapy, and a much smaller number of patients with cutaneous [15,16,17,18] or extra-cutaneous [19] melanomas. Currently, two BNCT drugs are available for clinical investigation: l-*para*-boronophenylalanine (BPA) and sodium mercaptoundecahydro-*closo*-dodecaborate (BSH), which is a derivative designed for brain tumor treatment. Despite their clinical use, both BPA and BSH show low selectivity, and great efforts have been made by several research groups to develop new and more selective boron delivery agents [20,21,22,23,24,25]. However, none of these have reached the stage where there is enough convincing data to warrant clinical biodistribution studies. More effective boron-containing agents are required so that they can be used alone or in combination with other agents to deliver the necessary amount of boron to cancer cells.

Human serum albumin (HSA) is one of the most suitable drug carriers [26,27,28,29,30,31,32]. Since the FDA approved Abraxane^®^ (paclitaxel-encapsulated albumin formulation), HSA has attracted increasing attention for therapeutic applications. It offers advantages of biocompatibility, low toxicity, and versatility as it contains a number of accessible functional groups for conjugation with low-molecular-weight compounds including anticancer drugs. In addition, serum albumin can transport the bound molecules to a specific destination and allow for a controlled release of its cargo known as ‘spatially controlled release’ [26,27,28,29,30,31]. This is achieved due to albumin’s interaction with specific receptors overexpressed in cancerous cells, which helps to specifically deliver albumin-bound molecules to tumor cells. Such receptors include glycoproteins Gp18, Gp30, and Gp60, as well as secreted protein acidic and rich in cysteine (SPARC) [33,34,35,36]. For example, the interaction of albumin molecules carrying paclitaxel with SPARC was shown to enable the increased local concentration of the drug released around the tumor cells. Likewise, the conjugation of boron-containing drugs with serum albumin would not only prolong the half-life of the drugs but also allow for the drug accumulation at the targeted tumor site [37,38,39,40]. Therefore, we chose serum albumin as a carrier for boron-based conjugates as potential anticancer therapeutics.

In order for BNCT to become a viable therapeutic option, the radiation dose delivered to the tumor must exceed the background radiation that healthy tissue receives from nonspecific neutron absorption [20]. As BNCT relies on neutron irradiation of the tumor-accumulated boron compounds for the therapeutic effect to occur, it is important to monitor the drug distribution inside the body in order to determine the optimal time window for the irradiation to be performed after drug administration. This can be achieved by introducing a tracking tag into the drug carrier. Examples of such tags include positron or gamma emitters that are incorporated into a nanoparticle-based drug delivering system. In this case, the drug distribution and localization can be monitored using positron emission tomography (PET) or single-photon emission computerized tomography (SPECT) imaging [41,42]. ^18^F-BPA positron emission tomography (PET) imaging [43,44] is now a well-established technique used as part of the treatment planning protocols both in Japan and Finland, the two countries where the largest number of patients have been treated by BNCT.

The emergence of hybrid scanners that allow for the simultaneous use of multiple imaging techniques (e.g., PET-CT (computed tomography), SPECT-CT, Optical-CT, MRI (magnetic resonance imaging)-PET, and MRI-Optical [45,46]) has advanced the development of multimodal imaging probes. Examples of such probes include bimodal probes, in which nuclear imaging techniques are combined with radio (CT, X-ray) or MR or optical imaging approaches [45]. At the same time, nuclear imaging requires ionizing radiation, which is hazardous and often short-lived. Therefore, alternative strategies for multimodal imaging are in demand.

^1^H and ^19^F provide very sensitive nuclei for MRI [45]. Among many detection modalities, ^19^F MRI is advantageous for deep-tissue and noninvasive imaging in vivo [47,48,49,50]. In vivo experiments using the C6 rat glioma model demonstrated that ^19^F MRI in combination with ^1^H MRI can selectively map the biodistribution of BPA labeled with an ^19^F atom (^19^F-BPA) [50]. The advantages of using the ^19^F nucleus include 100% abundance of the isotope and its high NMR sensitivity, which constitute 83% of that for a commonly used NMR nucleus—^1^H. Moreover, our body lacks fluorinated compounds present at physiological concentrations high enough to be detected with ^19^F MRI (a typical detection limit is less than 10^−3^ µmol/g wet tissue weight) [47]. As a result, a low endogenous background enables externally introduced fluorinated anticancer agents to be detected using ^19^F MRI with high contrast-to-noise ratio and specificity [48,49].

MRI advantages can be further enhanced if combined with optical imaging (OFI) [45,51]. Such a combination enables more detailed 3D information regarding the anatomy of the body in addition to selective and sensitive information regarding biodistribution of the administered agents available in real-time. At the same time, it is challenging to develop MRI/optical bimodal probes characterized by a similar level of sensitivity for their magnetic and optical constituents. A general strategy is to use the MRI probe in excess over the optical probe. Using such a multifunctional protein as human serum albumin offers a potential solution to that challenge.

The amine and sulfhydryl groups in albumin could be used for HSA modification [52,53,54]. The site-directed fluorophore labeling of albumin can be performed on the single free thiol group at cysteine 34 [48,52]. There are as many as 59 lysine residues in albumin [53], providing 59 amine groups as potential modification sites by ^19^F MRI labels [48,54,55]. Earlier, our group synthesized a fluorinated *N*-trifluroacetyl homocysteine thiolactone (HTLAc) [48] that readily and irreversibly reacts with *ε-*amino groups of albumin’s lysine residues. Three copies of trifluoroacetate and a single copy of a fluorophore Cy5 were covalently attached to a protein via suitable amino acids. For the synthesis of an albumin-based theranostic agent HSA-Cy5-HcyTFAc-B_12_H_11_, we used the reactivity of a thiolactone (a cyclic thioester) as a latent thiol functionality in thiol-‘click’ chemistry. The thiol was released by nucleophilic ring-opening (aminolysis) by amino groups on the HSA and subsequently reacted with a thiol ‘scavenger’ (a maleimide derivative of the undecahydro-*closo*-dodecaborate) [56] (Figure 1, arrow c).

Albumin also has 24 arginines [53] that, along with the lysines, could potentially be involved in the formation of a bimodal albumin-conjugate. The modification of arginine residues using dicarbonyl compounds is a common method to identify functional or reactive arginine residues in proteins [57]. In this work, we suggest the use of thenoyltrifluoroacetone (TTFA) as a part of the anticancer theranostic construct HSA-Cy5-HcyAc-B_12_H_11_-TTFA (Figure 1, arrow d). A trifluoromethyl moiety of the construct brings several advantageous properties, including narrow NMR signal due to the free rotation of the group even within high-molecular complexes, single signal for the three fluorine atoms, which allows one to avoid artifacts, and the absence of fluorine signals splitting on ^1^H nuclei in proximity. The latter property is particularly important as it prevents the loss of sensitivity and heating of biological samples to implement ^1^H decoupling in the ^19^F-detected spectrum.

Along with its beneficial properties as an imaging agent, TTFA is a promising chemotherapeutic agent. TTFA, an inhibitor of mitochondrial electron transport chain complex II, prevented the effects of hyperglycemia [58]. Hyperglycemia has been shown to promote cell proliferation, and tumor invasion and migration, along with increased chances of drug-resistance, which results in increased cancer prevalence and mortality [59]. By conjugating TTFA with albumin, the therapeutic index of the drug can be greatly improved, as the protein protects TTFA from renal clearance and, therefore, prolongs its circulation half-life. In addition, the enhanced permeation and retention effect of the macromolecular drug will allow preferential accumulation of the drug in tumor tissues. It will contribute to the local drug uptake. Therefore, we have proposed that the value of HSA as a BNCT drugs carrier may significantly increase if the TTFA-labeled homocystamide conjugate of human serum albumin is used for targeting. An HSA-based multidrug delivery system may represent an innovative delivery method for cancer therapeutics. As single drug-based therapeutic approaches may be compromised due to the mechanisms of intrinsic or acquired drug resistance [60,61], combination therapy allowing for a synergistic action of multiple anticancer agents via different signaling pathways appears as a potential solution [62,63].

Below, the fluorine-labeling procedures, spectroscopic and ^19^F NMR characteristics of the obtained boronate albumin conjugates, and comparative analysis of different homocystamide albumin conjugates for the glioma cells’ cytotoxicity are reported, together with preliminary examinations of biological behavior in vitro, such as boronated conjugate cytotoxicity before and after irradiation by epithermal neutron flux.

## 2. Results

### 2.1. Conjugation to Human Serum Albumin

Derivatives of the *closo*-dodecaborate anion are promising agents for boron BNCT [64]. Thus, of the multiple proposed strategies to introduce functional groups into the *closo-*dodecaborate anion, maleimide-dependent functionalization seems to be a promising one. Maleimide reacts with sulfhydryl groups of cysteine and homocysteine under mild conditions. This reaction has been used for the rapid assembly of a variety of albumin conjugates by so-called thiol-‘click’ chemistry [37,38,39,49,56,65]. For this reason, a maleimide derivative of the *closo-*dodecaborate anion (B_12_H_11_-mal) would be a potentially useful precursor for further direct conjugation of the drug to the carrier protein using a thiol-‘click’ chemistry.

HSA has 35 cysteine residues: 34 are paired in 17 disulfide bonds, leaving only Cys34 available for site-specific chemical modification [53]. Researchers [37,38,39] have developed maleimide-containing *closo-*dodecaborate (MID) and aimed to conjugate it to albumin at Cys34. However, MID was found to conjugate not only to free SH of cysteine residue but also to lysine residues in albumin. As maleimide bioconjugation chemistry has been essentially linked with the construction of well-defined therapeutics, issues related with lysine cross-reactivity and heterogeneous preparations of albumin-drug conjugates have inspired us to develop a next generation of maleimido-type Michael acceptors.

For the synthesis of the albumin-based HSA-Cy5-HcyTFAc-B_12_H_11_, we first labeled HSA with a fluorophore Cy5, whose strong fluorescence would enable the monitoring of internalization of the HSA conjugates into cells [48]. For the labeling strategy, a conventional maleimide chemistry was employed to position the fluorophore at Cys34 of the protein (Figure 1, arrow a). Next, we used the reactivity of a thiolactone (a cyclic thioester) as a latent thiol functionality in the thiol-‘click’ chemistry to establish additional SH groups into the protein conjugate. The use of the trifluoroacetyl derivative of homocysteine thiolactone (HTLTFAc) at this stage provided a convenient route to introduce fluorine labels into the conjugate (Figure 1, arrow b). Ellman’s test [66] showed the incorporation of 3.0 ± 0.1 sulfhydryl groups per protein molecule upon reacting HSA-Cy5 with HTLTFAc. Based on MALDI-TOF/TOF mass spectrometry analysis of the tryptic fragments of HSA-Cy5-HcyTFAc, the HcyTFAc residues were attached to Lys-199, Lys-414, and Lys-557/560 of HSA (Appendix A).

The thiol was released via nucleophilic ring-opening (aminolysis) by amino groups on HSA and subsequently reacted with a thiol ‘scavenger’ (a maleimide derivative of the drug), as depicted in Figure 1, arrow c. All introduced SH groups were subsequently modified with B_12_H_11_-mal, as no free sulfhydryl groups were detected in the Ellman’s test after the conjugation was complete.

It might be simply considered that the sensitivity in ^19^F NMR is easily improved by increasing the number of ^19^F nuclei in the probe. Albumin has 24 arginines [53] that, along with the lysines, could also potentially be involved in the formation of the bimodal albumin-conjugate. For the conjugate HSA-Cy5-Hcy-Ac-B_12_H_11_-TTFA, the introduction of fluorine was carried out by modifying arginine residues using TTFA (Figure 1, arrow d). The use of TTFA for the fluorination of albumin makes it possible to introduce almost twice as much fluorine into the protein structure as in the case of using HTLTFAc for this purpose. The presence of TTFA residue in the final conjugate was proven by the appearance of the characteristic maximum at 337 nm (Figure 2A). It was calculated that the constant absorbance value corresponded to an albumin preparation, which contained 5.4 TTFA residues per protein molecule.

The fluorine-labeled albumin conjugates were further characterized by ^19^F NMR. The ^19^F NMR spectra of the fluorinated conjugates are given in Figure 2B. The HSA-Cy5-HcyTFAc-B_12_H_11_ and HSA-Cy5-HcyAc-B_12_H_11_-TTFA conjugates exhibited ^19^F signals at ca. 88 ppm.

The *N*-Hcy-HSA is more susceptible to oxidation than is HSA and the number of aggregates increases [67]. It was shown [48] that total oligomers increased to 83% in the samples of albumin containing unmodified *N*-homocysteine residues. At the same time, our results indicated that the blocking of the alpha-amino group of HTL can inhibit the aggregation of *N*-homocysteinylated HSA. Our HSA-Cy5-HcyTFAc-B_12_H_11_ conjugate was 56.4% monomeric with *M_W_* ~66.5 kDa and 39.6% oligomers, as shown by SDS-PAGE (Appendix A, lane 2, and Appendix A). The modification can also affect protein conformation [48]. In order to evaluate the conjugation effect on the HSA secondary structure, we analyzed far-UV CD spectra of the intact and modified protein (Appendix A). Our results clearly demonstrated that the boronated albumin-based theranostic agent retained most of the α-helices present in the native protein (Appendix A).

Surprisingly, *N*-substituted HTL (HTLAc) appears to be a suitable starting material for medicinal chemistry. Its Hcy conjugate HSA-Cy5-HcyAc-B_12_H_11_-TTFA has been demonstrated to have beneficial physicochemical properties. We found that total oligomers decreased to 10% in the samples of albumin containing modified HcyAc residues (Appendix A, line 4, Appendix A). *N*-acetyl HTL is a well-known protein-thiolating agent [68,69,70,71] and a drug used in liver therapy under the name citiolone due to its radical-scavenging properties. Despite the wide use of *N*-acetyl HTL, the cytotoxicity of its protein conjugates is not very well known. Therefore, more detailed studies with *N*-substituted albumin homocystamide conjugates involving toxicity, metabolic rate, and immunologic consequences are necessary to establish their potential as multimodal probes for molecular imaging and anticancer therapy.

### 2.2. In Vitro Studies

The effect of HSA-Cy5-HcyTFAc-B_12_H_11_ and HSA-Cy5-HcyAc-B_12_H_11_-TTFA conjugates on the viability of the cancer cells was determined by the standard colorimetric MTT assay [72] using T98G cells. In the absence of neutron irradiation, the cell line retained a proliferation rate of over 80% upon treatment with the boronated conjugate within its concentration range of 0.02–30 μM (Figure 3). Thus, for neutron source efficacy evaluation, ~30 μM of boronated albumin can be used to minimize the effect of the drug on glioma cell colony formation.

The intracellular uptake, and accumulation and distribution of the boronated albumin theranostic conjugate were investigated by flow cytometry and confocal microscopy. The Cy5 fluorescent label (emission at 699 nm upon excitation at 633 nm) was incorporated into the boron-albumin conjugate to enable monitoring of the conjugate in cells as far-red fluorescence does not overlap with cellular autofluorescence. Flow rate analysis showed that 98% of the T98G cells incubated with HSA-Cy5-HcyTFAc-B_12_H_11_ (20 μM) for 2 h accumulated the fluorescently labeled HSA derivative (Figure 4). The detailed kinetics of accumulation was not examined. At the same time, relative to the untreated cell control (left panel), ~6% of cells were observed as fluorescent in the ‘0 h’ control, in which the cells were exposed to the HSA conjugate for ~1 min (middle panel), thus speaking in favor of the rapid cellular uptake of the conjugate.

For confocal microscopy analysis, the T98G cell line was incubated with the HSA-Cy5-HcyTFAc-B_12_H_11_ conjugate (20 μM) for 1.5 h. The confocal microscopy images (Figure 5) showed the presence of the HSA-Cy5-HcyTFAc-B_12_H_11_ conjugate as punctuated small vesicles distributed in the cytoplasm of the T98G cells, suggesting an endocytic internalization mechanism [73,74]. Furthermore, orthogonal projections of z-stack acquisitions proved that vesicles and large endocytic structures were indeed present inside the treated cells (Figure 6).

### 2.3. Neutron Irradiation Experiments

Recently, several accelerators destined for hospital placement have been introduced [75]. For BNCT purposes, a proton accelerator with vacuum insulation and a lithium target have been developed at the Budker Institute of Nuclear Physics (BINP) at the Russian Academy of Sciences (Novosibirsk, Russian Federation) [76]. To date, initial radiobiological experiments on tumor cells to evaluate the efficacy of the neutron source at BINP have been performed with l-*p*-boronophenylalanine (BPA) [77,78], a boron agent. In the current article, we provide an in vitro efficacy evaluation of this unique accelerator-based neutron source, for boron neutron capture therapy (BNCT) at BINP, using BPA and our albumin conjugate. We believe that our study will bring small clarity to the ongoing in vitro experiments on neutron capture therapy and help other researchers to advance accelerator-based BNCT into the clinical phase.

The glioma cells were incubated in medium with HSA-Cy5-HcyTFAc-B_12_H_11_ conjugate or with BPA and irradiated by an epithermal neutron flux. Boron-negative cells, irradiated by neutrons, were used as controls. Survival of the U87MG tumor cells incubated in medium with the HSA-Cy5-HcyTFAc-B_12_H_11_ conjugate or with BPA reduced with the increase of neutron flux (Figure 7). Survival data were fit by a linear regression according to the linear-quadratic formula (LQ fit) [79]. We did not receive any clones after the irradiation of U87MG cells with pretreatment BPA. Although the increase in neutron flux affected the survival of the cells without boron, pretreatment with HSA-Cy5-HcyTFAc-B_12_H_11_ or BPA resulted in a significantly lower fraction of the survived U87MG cells, as compared with the conjugate-free cells (Cntr) (*p* < 0.0001). Multiple comparisons by two-way ANOVA of the survived fractions did not show a statistical difference between the U87MG cell line pretreated with HSA-Cy5-HcyTFAc-B_12_H_11_ and BPA.

The treatment with the HSA-Cy5-HcyTFAc-B_12_H_11_ conjugate and BPA before BCNT inhibited the proliferation of the U87MG cell line in a time-dependent manner (Figure 8). The colony-forming assay showed that a neutron flux of 4 × 10^12^ cm^−2^ did not significantly affect cell viability without boron, but was effective in treating the tumor cells with boron accumulated. Our cell survival data confirmed the efficacy of the accelerator neutron source with the lithium target at BINP to produce a sufficient number of neutrons to initiate a boron neutron capture reaction within and in proximity to tumor cells. Decreasing the integral of the proton current to 3 *mA × h* can mitigate the slight effect of irradiation on the control cells observed in some of the experiments.

We performed initial in vitro experiments to evaluate the efficacy of the obtained conjugates. These data will be supplemented with the data obtained using appropriate animal models as a continuation of this study.

## 3. Materials and Methods

### 3.1. Chemicals, Reagents, Cancer Cells, and Facilities

Human serum albumin (HSA) was obtained from Sigma–Aldrich Chem. Co. (St. Louis, MO, USA). The product number of HSA used was A3782. The SH contents of albumin and albumin products were determined using the Ellman’s method as described in the literature at pH 8, and employed DTNB (5,5′-dithio-bis(2-nitrobenzoic acid) spectrophotometrically at 412 nm (*ε* = 1.36 × 10^4^ M^−1^cm^−1^) [66]. The concentrations of albumin solutions were determined by absorption at 292 nm, pH 13, using the molar extinction coefficient *ε* = 4.44 × 10^4^ M^−1^cm^−1^ [53].

Reagents and materials were purchased from Sigma-Aldrich (St. Louis, MO, USA), unless otherwise indicated. MTT (3-[4,5-dimethylthiazol-2-yl]-2,5-diphenyltetrazolium bromide) assay kit was purchased from Invitrogen (Waltham, MA, USA). Milli-Q water with a conductivity greater than 18 MΩ/cm was used in all experiments. Phosphate-buffered saline (PBS) (0.01 M, pH 7.3–7.5, Biolot) was used.

Human glioma cell lines: T98G and U87MG cells, were obtained from the Russian cell cultures collection (Russian Branch of ETCS, St. Petersburg, Russia). T98G cells were cultured in IMDM and DMEM growth media supplemented with 10% fetal calf serum (FCS), penicillin (100,000 IU/L), streptomycin (100,000 IU/L), and amphotericin B (1 mg/L) in a humidified atmosphere containing 5% CO_2_ at 37 °C. U87MG cells were cultured at 37 °C in 5% CO_2_ in α-MEM containing 10% fetal bovine serum (FBS), 1 mM of sodium pyruvate, 50 U/mL of penicillin, and 100 g/mL of streptomycin.

Electronic absorption spectra were acquired on a UV-1800 spectrometer (Shimadzu, Kyoto, Japan).

^19^F NMR spectra were recorded on an AV-300 NMR spectrometer (Bruker, Billerica, MA, USA) at 282.7 MHz. The spectra were detected at 25 °C in 5 mm NMR sample tubes. C_6_F_6_ (*δ* 0.00 ppm) was used as an external reference for chemical shifts in ^19^F NMR spectra. Chemical shifts (*δ*) are reported in parts per million (ppm).

Inductively coupled plasma–atomic emission spectroscopy was performed using an ICPE-9820 (Shimadzu, Kyoto, Japan). A conjugate sample in PBS (40 µL, 0.4 mM) was diluted to 4 mL with double-ionized water and used for measurement.

Confocal microscopy analysis was performed using an LSM 780 NLO (Zeiss, Oberkochen, Germany) on the basis of AxioObserver Z1 (Zeiss) at the Microscopy Center of the Institute of Cytology and Genetics, SB RAS, Russia.

### 3.2. Synthesis and Characterization of Multifunctional Human Serum Albumin-Therapeutic Conjugates

The maleimide-conjugating *closo-*dodecaborate tetramethylammonium form (B_12_H_11_-mal) was synthesized according to the published procedures [37,80] (see Appendix A).

The synthetic procedure for HSA-Cy5-HcyTFAc and HSA-Cy5-HcyAc conjugates was adapted from Chubarov et al. [48]. Detailed synthetic procedures are given in Appendix A.

#### 3.2.1. Synthesis and Characterization of Theranostic Conjugate HSA-Cy5-HcyTFAc-B_12_H_11_

A solution (0.5 mL, 0.7 mM, 0.35 μmol) of the HSA-Cy5-HcyTFAc in PBS buffer (pH 7.4) was mixed with B_12_H_11_-mal in DMSO (29.2 μL, 1.58 μmol, 0.166 M). The reaction mixture was incubated under constant gentle stirring at 37 °C in the dark for 17 h. The protein conjugate was purified by SEC utilizing a Millipore ultrafiltration tube and stored at 4 °C. The yield of the HSA-Cy5-HcyTFAc-B_12_H_11_ derivative was ~65%. UV-vis (PBS buffer, pH 7.4): *λ*_max_ 278 nm (*ε* = (3.88 ± 0.1) × 10^4^), *λ*_max_ 646 nm (ε = (4.99 ± 0.1) × 10^4^). ^19^F NMR (PBS buffer, pH 7.4, d, ppm): ~88 ppm (CF_3_, TFAc residues). Inductively coupled plasma–atomic emission spectroscopy: 2.8 B_12_H_11_ residues per albumin.

The molecular weight of the conjugate was determined by MALDI-TOF analysis, and differences in *MW* were used to calculate the number of B_12_H_11_ bound per albumin. The HSA-Cy5-HcyTFAc-B_12_H_11_ conjugate had a calculated molecular mass of 68.203 kDa, and 34.101 kDa for the double-charged protein (Appendix A). The difference between the homocysteinylated and native species is 2524 Da, which corresponds to three *N*-homocysteinyl-B_12_H_11_ moieties, *N*-linked by amide linkages to Lys of HSA (*N*-Lys-HcyTFAc-B_12_H_11_; 586 Da) plus one dye moiety on Cys34 (S-Cys-Cy5; 766.4 Da).

#### 3.2.2. Synthesis and Characterization of Multifunctional Human Serum Albumin-Therapeutic Conjugate HAS-Cy5-HcyAc-B_12_H_11_-TTFA

The HSA-Cy5-HcyAc-B_12_H_11_ synthesis was carried out in the same way as HSA-Cy5-HcyTFAc-B_12_H_11_ synthesis. The yield of the HSA-Cy5-HcyAc-B_12_H_11_ derivative was ~65%. UV-vis (PBS buffer, pH 7.4): *λ*_max_ 278 nm (*ε* = (3.88 ± 0.1) × 10^4^), *λ*_max_ 646 nm (ε = (4.99 ± 0.1) × 10^4^). Inductively coupled plasma–atomic emission spectroscopy revealed 1.8 B_12_H_11_ residue connections to an albumin molecule. The HSA-Cy5-HcyAc-B_12_H_11_ conjugate had a calculated molecular mass of 67.250 kDa, and 33.625 kDa for the double-charged protein (Appendix A). The difference between the homocysteinylated and native species is 1721 Da, which corresponds to two *N*-homocysteinyl-B_12_H_11_ moieties, *N*-linked by amide linkages to Lys of HSA (*N*-Lys-HcyAc-B_12_H_11_; 532 Da) plus one dye moiety on Cys34 (S-Cys-Cy5; 766.4 Da.

A solution (0.213 mL, 0.71 mM, 0.15 μmol) of the HSA-Cy5-HcyAc-B_12_H_11_ derivative in 0.1 M of borate buffer (pH 10.5) was mixed with TTFA in DMSO (10.65 μL, 1.68 μmol, 0.373 mg). The reaction mixture was incubated under constant gentle stirring at 37 °C in the dark for 8 h. The protein conjugate was purified by SEC utilizing a Millipore ultrafiltration tube and stored at 4 °C. The yield of HSA-Cy5-HcyAc-B_12_H_11_-TTFA was ~95.6 %. UV-vis (PBS buffer, pH 7.4): λ_max_ 278 nm (ε = (3.88 ± 0.1) × 10^4^), λ_max_ = 337 nm (ε = (2.17 ± 0.1) × 10^4^), λ_max_ 650 nm (ε = (4.99 ± 0.1) × 10^4^). ^19^F NMR (PBS + D_2_O, δ, ppm): 88.60 (s, CF_3_). Inductively coupled plasma–atomic emission spectroscopy revealed 30 boron atoms bound to an albumin molecule.

### 3.3. Cell Viability Assay (MTT Test)

Cytotoxicity assays were performed using the MTT test, [72]. For this purpose, cells were grown to the exponential growth phase and seeded in 96-well plates to achieve a cell concentration of 2000 cells per well. The cells were incubated for 24 h prior to their treatment with the medium containing albumin conjugates to achieve HSA-equivalent concentrations ranging from 0.02 to 60 µM. The treatment of the cells with HSA derivatives was carried out for 72 h at 37 °C, after which MTT was added to achieve a 0.5 mg/mL final concentration. After incubation at 37 °C for 2 h, the medium was removed, and 100 μL of isopropanol was added to each well to dissolve formazan crystals. The plate was analyzed using a microplate reader Multiscan FC (Thermo Fisher Scientific Corporation) at the absorbance peak at 570 nm with the absorbance at 620 nm used as a baseline. The analysis of three independent experiments was carried out. The data are shown as mean values with standard deviations.

### 3.4. Intracellular Distribution of Boronated Albumin Theranostic In Vitro

T98G cells (10^5^/mL, 100 µL) were seeded in Thermo Scientific™ Nunc™ MicroWell™ 96-Well Optical-Bottom Plates with a Coverglass Base, in IMDM containing 10% FBS, penicillin, and streptomycin and were preincubated for 17 h in humidified atmosphere with 5% CO_2_. The medium was exchanged for the fresh one containing 20 μM of HSA-Cy5-HcyTFAc-B_12_H_11_. The cells were incubated at 37 °C for 1.5 h. Time-lapse images were sampled every 5 min over a period of 1.5 h on a LSM780 microscope (Zeiss, Oberkochen, Germany). Red fluorescence of Cy5 was detected at 699 nm upon excitation at 633 nm. Then, the cells were washed 3 times with PBS and the intracellular localization of HSA was assessed by scanning fluorescence microscopy. Then, cells were fixed in 4% formaldehyde for 20 min at room temperature and counterstained by SYBR Green 1 (1:10,000). The intracellular distribution of HSA and SYBR Green 1-stained nuclei was observed by scanning fluorescence microscopy. Green fluorescence of SYBR Green 1 was detected at 571 nm upon excitation at 488 nm (band of Ar laser). The images were processed in the ZEN Blue light program.

### 3.5. Flow Cytometry

Flow cytometry was used to characterize the levels of intracellular accumulation of HSA-Cy5-HcyTFAc-B_12_H_11_ in T98G cells after 0 and 2 h of incubation. A number of 1 × 10^6^ cells per well were seeded in 12-well plates and then detached by adding trypsin (2%) (MP Biomedicals, Irvine, CA, USA). The cells were resuspended in DMEM, pelleted by centrifugation at 200 *g* for 5 min, washed with PBS, and fixed with 2% formaldehyde in PBS buffer for 10 min at room temperature. The fixed cells were analyzed using a NovoCyte flow cytometer (ACEA Biosciences, Santa Clara, CA, USA) with the data processed using NovoExpress software (ACEA Biosciences, Santa Clara, CA, USA). Three independent experiments were performed for each of the conditions analyzed. Intracellular accumulation of albumin’s derivatives was characterized by the percentage of fluorescently labeled cells and the mean fluorescence of the cells in a sample.

### 3.6. Neutron Irradiation Experiments

Neutron irradiation was carried out at Budker Institute of Nuclear Physics (Novosibirsk, Russia), using the accelerator-based neutron source [76]. After 18 h of incubation with the HSA-Cy5-HcyTFAc-B_12_H_11_ conjugate (31 μM) or BPA-containing boron-10 (20 ppm, 10 μg/mL), human glioma cells were irradiated by epithermal neutron flux over 2 h. Boron-negative cells, irradiated by neutrons, were used as controls. The irradiation conditions were as follows: proton energy of 2.1 MeV, proton current of 1.5–3.0 mA to enable an epithermal neutron flux up to 3 × 10^8^ cm^−2^ s^−1^. The neutron flux was measured by a gold foil activation technique using a detector with a lithium-containing scintillator (GS20, Saint-Gobain Crystals, Hiram, OH, USA). The neutron flux was set up at 4 × 10^12^ cm^−2^ and 8 × 10^12^ cm^−2^.

The cytotoxicity of the HSA-Cy5-HcyTFAc-B_12_H_11_ conjugate in BNCT was evaluated by colony forming assay [79]. After neutron irradiation, the cells were incubated at 37 °C in a 5% CO_2_ atmosphere for 8 days, fixed with glutaraldehyde, and stained with crystal violet, and colonies of more than 50 cells were counted. A cell proliferation assay was also performed after irradiation using an MTT assay. All cells were seeded in 96-well plates at a density of 1000 cells per well after irradiation. After 2, 4, or 6 days of incubation, proliferation of U87MG cells was assessed using a 3-(4,5-dimethylthiazol-2-yl)-2,5-diphenyl tetrazolium bromide (MTT) assay. The optical density was recorded using a microplate reader (Apollo LB 912, Berthold Technologies, Bad Wildbad, Deutschland) at 570 nm, with a reference wavelength of 620 nm. Two-way ANOVA was used for comparisons of more than two sets of data. Differences were considered to be significant if the *p*-value was <0.05.

## 4. Conclusions

Boron-albumin conjugates are promising anti-cancer therapeutics as they can be administered less frequently than alternative therapeutic agents and yet offer sufficient accumulation of the action component at the tumor site, which can improve the quality of life for cancer patients. We report successful preparation of a new albumin-based boron theranostic agents with multimodal functions, such as fluorescence imaging (Cy5) and ^19^F MRI (*N*-trifluoroacetylhomocysteine, or thenoyltrifluoroacetone residue). The use of TTFA for the fluorination of albumin makes it possible to introduce almost twice as much fluorine into the protein structure as in the case of using HTLTFAc for this purpose. We proposed that the value of HSA as a BNCT drugs carrier may significantly increase if the TTFA-labeled homocystamide conjugate of human serum albumin is used for targeting. Along with its beneficial properties as an imaging agent, TTFA is a promising chemotherapeutic agent. An HSA-based multidrug delivery system may represent an innovative delivery method for cancer therapeutics. The conjugation of albumin with undecahydro-*closo*-dodecaborate did not significantly affect cell viability in the absence of irradiation, as compared with the unmodified protein. However, neutron capture by this boron-containing albumin decreased the tumor cell survival. Conjugation of the boron-based drug to HSA—a carrier protein with a long plasma half-life—is expected to extend its systemic circulation and preserve its activity. The presence of fluoro-organic residues and a single copy of a fluorophore Cy5 will enable the monitoring of the drug distribution by two different modes, thus making the reported HSA conjugates a real theranostic tool.

## Figures and Tables

**Figure 1 molecules-26-06537-f001:**
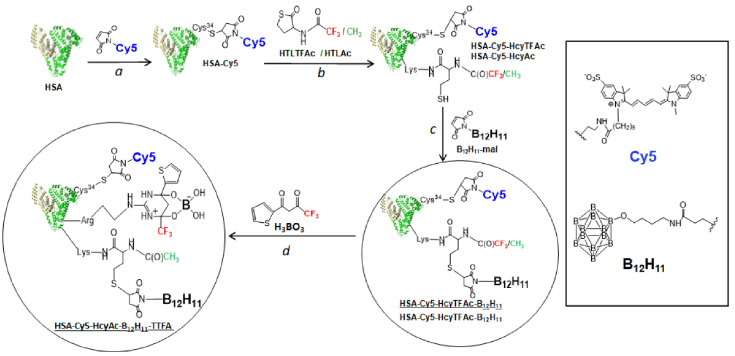
Synthetic routes to obtain the multifunctionalized serum albumin architecture—HSA-Cy5-HcyTFAc-B_12_H_11_ and HSA-Cy5-HcyAc-B_12_H_11_-TTFA. Drug carrier (shown schematically as a heart-like structure)—human serum albumin (HSA). Effector—B_12_H_11_: therapeutic agent. Note that homocysteine thiolactone derivatives (HTLTFAc and HTLAc) are used as a functional handle. HTLTFAc and TTFA are used as a source of fluorine atoms. Optical imaging—fluorescent dye Cy5 conjugated with Cys-34.

**Figure 2 molecules-26-06537-f002:**
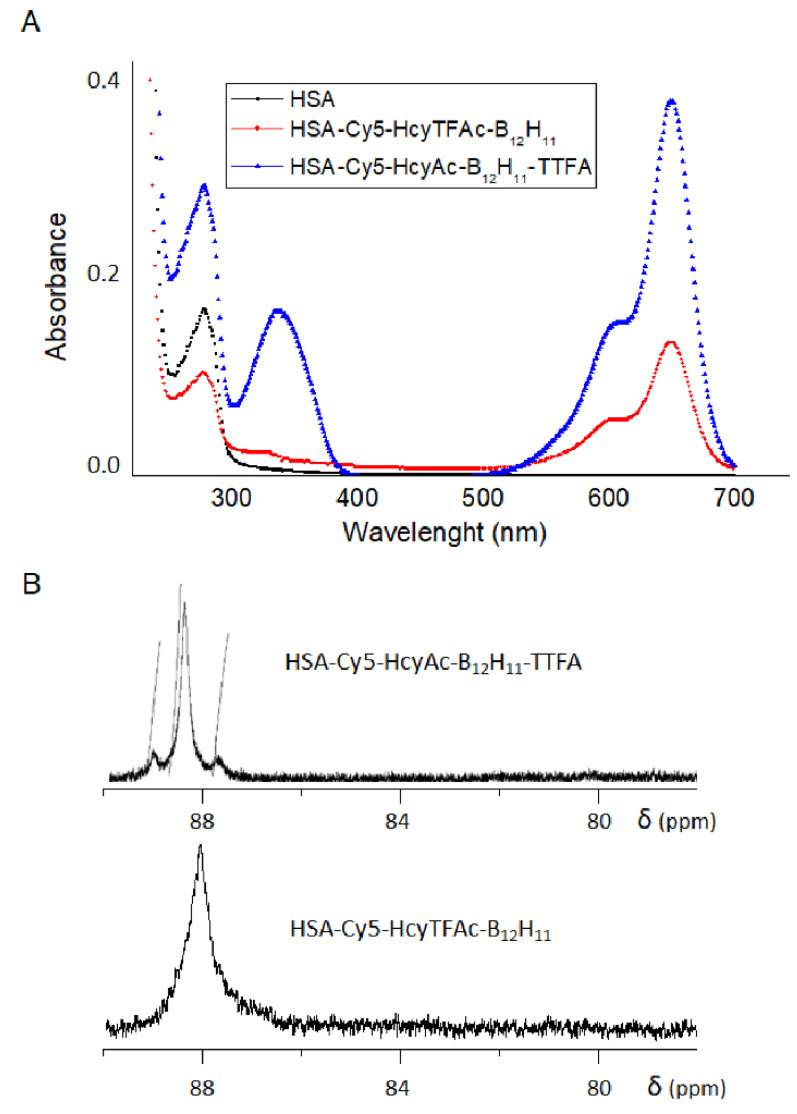
Characteristics of multifunctional human serum albumin conjugates. (**A**) UV-vis spectra of HSA and boronated albumin theranostic conjugates in PBS buffer, pH 7.4. HAS—black; HSA-Cy5-HcyTFAc-B_12_H_11_—red; HSA-Cy5-HcyAc-B_12_H_11_-TTFA—blue. (**B**) ^19^F NMR spectrum (at 282.4 MHz) of HSA-Cy5-HcyAc-B_12_H_11_-TTFA (0.6 mM) and HSA-Cy5-HcyTFAc-B_12_H_11_ (0.3 mM) in PBS buffer (pH 7.4; to provide deuterium lock, D_2_O was added to 20% of the total volume) at 37 °C. The chemical shifts are referred to the resonance of C_6_F_6_ at 0.0 ppm.

**Figure 3 molecules-26-06537-f003:**
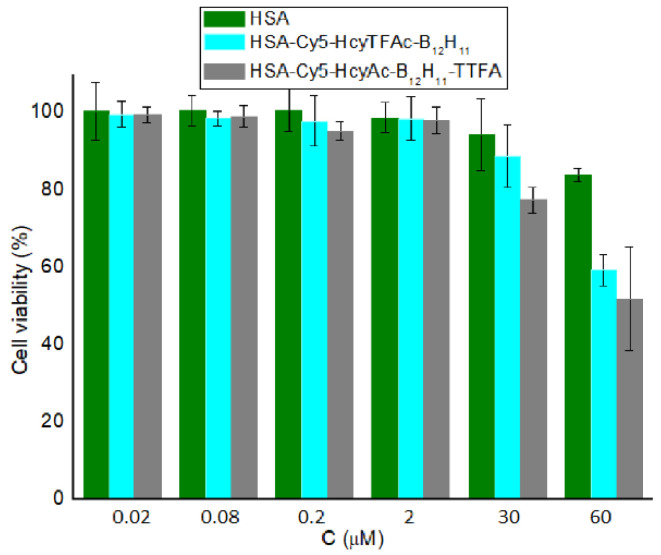
Viability of T98G cells. The cancer cell line was incubated for 72 h with HSA (green), HSA-Cy5-HcyTFAc-B_12_H_11_ (cyan), and HSA-Cy5-HcyAc-B_12_H_11_-TTFA (gray) at various concentrations. After incubation, the cell viability was measured using the MTT test. The reported values represent the mean ± SD (*n* = 3). Two-way ANOVA was used for comparisons of more than two sets of data. Differences were considered to be significant if the *p*-value was <0.05.

**Figure 4 molecules-26-06537-f004:**
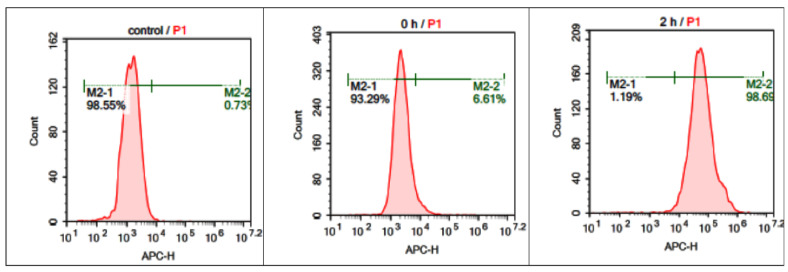
In vitro cellular uptake of HSA-Cy5-HcyTFAc-B_12_H_11_ by T98G cells measured by flow cytometry. Red color: FACS analysis; green color: percent cellular uptake of HSA-Cy5 and HSA-Cy5-HcyTFAc-B_12_H_11_. The data were normalized to nontreated cells (control).

**Figure 5 molecules-26-06537-f005:**
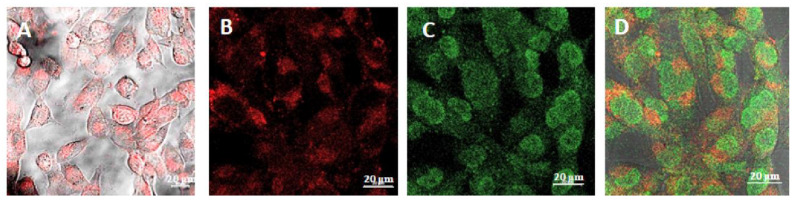
Representative images of confocal microscopy analysis of theT98G cell line treated with the fluorescent HSA-Cy5-HcyTFAc-B_12_H_11_ conjugate (20 µM) for 1.5 h. Cell nuclei were stained with SYBR Green I. HSA-Cy5-HcyTFAc-B_12_H_11_ is visible as a red color. Scale bars: 20 μm. Panel (**A**): Live cell image: T98G cells were incubated with the conjugate for 1.5 h and washed three times with PBS. Panels (**B**–**D**): The cells were fixed with 2% formaldehyde. Panel (**D**): Merged (**B**,**C**).

**Figure 6 molecules-26-06537-f006:**
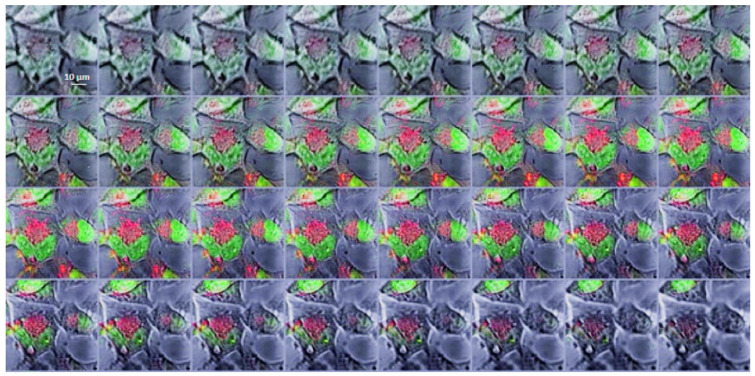
Gallery of merged images acquired along the z-axis of SYBR Green I-stained T98G cells treated with the HSA-Cy5-Hcy-TFAc-B_12_H_11_ conjugate. Cell nuclei are visible as a green color. HSA-Cy5-HcyTFAc-B_12_H_11_ is visible as a red color. Every subsequent image was taken 0.68 μm higher than the previous one. Scale bars: 10 μm.

**Figure 7 molecules-26-06537-f007:**
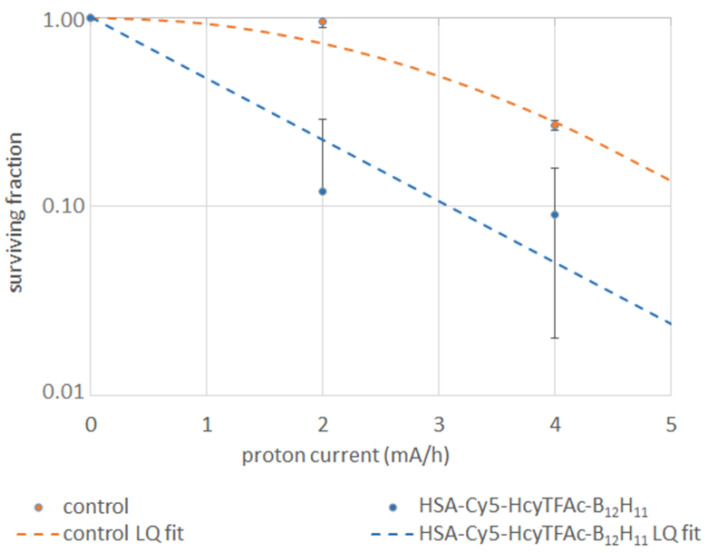
Survival curves of U87MG cells incubated with HSA-Cy5-HcyTFAc-B_12_H_11_ conjugate (cyan curve) and without reagents (orange curve) depending on neutron fluence.

**Figure 8 molecules-26-06537-f008:**
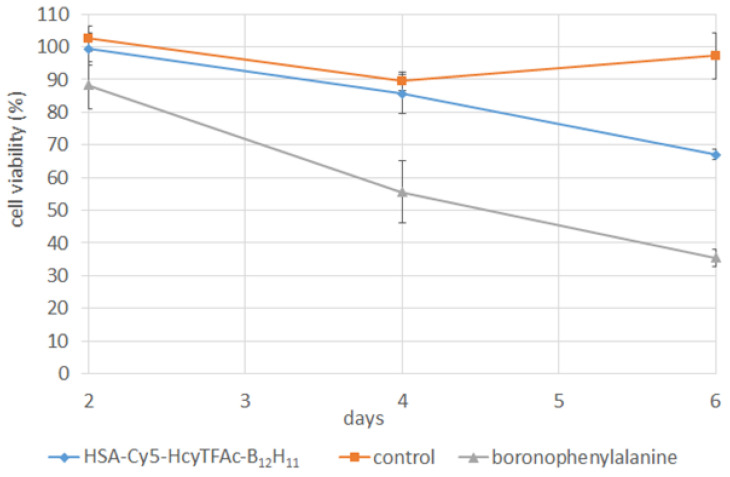
The cell viability of U87MG cells incubated with HSA-Cy5-HcyTFAc-B_12_H_11_ conjugate and BPA before BNCT at 2, 4, and 6 days after neutron irradiation. Control: U87MG cells after neutron irradiation without boron-containing compounds.

## Data Availability

Not applicable.

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
