# Peer review of "Homocystamide Conjugates of Human Serum Albumin as a Platform to Prepare Bimodal Multidrug Delivery Systems for Boron Neutron Capture Therapy"

_molecules, 2021, doi:10.3390/molecules26216537_

Round 1
Reviewer 1 Report
This is an interesting study about homocystamide conjugates of human serum albumin as a plat3 form to prepare bimodal multidrug delivery systems for boron4 neutron capture therapy. This study may ab accepted for publication with major revision
- In figure 3, the statistical analysis must be done to show the significance between different treatments, also need to mention standard deviation errors and p-value.
- Figure 5 A and 5D are not clear, needs to be replaced the better quality picture
- Figure 8, this graph is not clear, is it about cell inhibition or cell viability, because optical density is mentioned in, control sample there is an increase in the optical density which indicates what? please clarify and redraw the graph
- They need to show the response of conjugates also on normal and healthy cells
Author Response
The authors are grateful to Reviewer #1 for insightful comments on the manuscript. The authors are grateful to Reviewer #1 also for the expressed questions/criticism, it helped to improve our manuscript a lot. We have addressed the issues highlighted by the reviewer and made the appropriate changes in the manuscript in accordance with the reviewer’s comments and suggestions. We highlighted (yellow highlighted text) all changes made when revising the manuscript.
- In figure 3, the statistical analysis must be done to show the significance between different treatments, also need to mention standard deviation errors and p-value.
Our response, and changes made to the manuscript:
We have edited the text and Figure 3 in accordance with the conditions of our experiment (page 7, lines 254-255).
Figure 3. Viability of T98G cells. The cancer cell line was incubated for 72 h with HSA (green); HSA‑Cy5-HcyTFAc-B12H11 (cyan); HSA‑Cy5-HcyAc-B12H11-TTFA (gray) at various concentrations. After incubation, the cell viability was measured using the MTT test. The reported values represent the mean ± SD (n = 3). Two-way ANOVA was used for comparisons of more than two sets of data. Differences were considered to be significant if the P-value was <0.05.
- Figure 5 A and 5D are not clear, needs to be replaced the better quality picture
Our response, and changes made to the manuscript:
We have made the Figure 5 brighter and more contrasting.
- Figure 8, this graph is not clear, is it about cell inhibition or cell viability, because optical density is mentioned in, control sample there is an increase in the optical density which indicates what? please clarify and redraw the graph
Our response, and changes made to the manuscript:
We have changed the text to better reflect the findings of our work. The following text “Survival data were fit by a linear regression according to the linear-quadratic formula (LQ fit) [68]. We did not receive any clones after irradiation U87MG cells with pretreatment BPA.” has been added (page 9, lines 304-306):
“The glioma cells were incubated in medium with HSA-Cy5-HcyTFAc-B12H11 conjugate or with BPA and irradiated by epithermal neutron flux. Boron-negative cells, irradiated by neutrons were used as controls. Survival of U87MG tumor cells incubated in medium with HSA-Cy5-HcyTFAc-B12H11 conjugate or with BPA reduced with the increase of neutron flux (Figure 7). Survival data were fit by a linear regression according to the linear-quadratic formula (LQ fit) [68]. We did not receive any clones after irradiation U87MG cells with pretreatment BPA. Even though, the increase of neutron flux affected survival of the cells without boron, pretreatment with HSA-Cy5-HcyTFAc-B12H11 or BPA resulted in the significantly lower fraction of of the survived U87MG cells as compared with the conjugate-free cells (Cntr) (P < 0.0001). Multiple comparisons by two-way ANOVA of the survived fractions did not shown statistical difference between the U87MG cell line pretreated with HSA-Cy5-HcyTFAc-B12H11 and BPA.”
We have edited the text (page 10, lines 313-314 and 325-326) and Figures 7 and 8 in accordance with the conditions of our experiment.
Figure 7. Survival curves of U87MG cells incubated with HSA-Cy5-HcyTFAc-B12H11 conjugate (cyan curve) and without reagents (orange curve) depending on neutron fluence.
Figure 8. The cell viability of U87MG cells incubated with HSA-Cy5-HcyTFAc-B12H11 conjugate and BPA before BNCT at 2, 4, 6 days after neutron irradiation. Control: U87MG cells after neutron irradiation without boron-containing compounds.
- They need to show the response of conjugates also on normal and healthy cells
Our response:
Glioma is the most invasive brain tumor with a poor prognosis. There are several critical issues, as itemized below, that must be addressed before applying BNCT as a useful modality for cancers treatment.
- More effective boron-containing agents are required so that they can be used alone or in combination with other agents to deliver the necessary amount of boron to cancer cells.
- The delivery of boron-containing agents to the cancer cells and cellular microdistribution must be optimized to improve its uptake, especially to diverse subpopulation of cancer cells.
- The development of new imaging agents to provide estimates of boron content in the cancer cells are essential. Therefore, the development of new imaging agents is the other fundamental need for the future progress of BNCT.
The goal of our work is to develop a new drug for targeted delivery of boron for BNCT and for provided estimates of boron content in the cancer cells. We demonstrated intracellular uptake of the boron-albumin conjugate by the human glioma cells. The localization of albumin-based boron theranostic agent was confirmed by fluorescence microscopy. In Figs. 5 and 6 show the distribution of HSA conjugate in tumor cells. These results are fully consistent with the design expectations and provide experimental support for the suggestion that homocystamides HSA are attractive candidates for use in boron compounds delivery systems in the human glioma cells. Although the uptake mechanism of albumin by the cells is still not clear except for gp60-mediated transcytosis of endothelial cells (J. E. Schnitzer and P. Oh, 1994) and SPARC facilitated albumin accumulation (N. Desai, et al., 2004), an interesting in vivo study showing evidence of albumin homocystamides uptake has been recently reported (V.A. Lisitskiy, et al., 2017). Lisitskiy et al. found that malignant glioma actively scavenged extracellular proteins, such as albumin conjugates. Our current investigation of localisation of boron-albumin conjugate in the human glioma cells also supports their results. Herein, it was extremely important for us to compare its effect with the effect of the only approved drug, boron-phenylalanine. This drug is not yet final, since we used natural boron with 20% boron-10 content, and not enriched to 98%. At this stage of the research, it was important for us to compare these two drugs under the same irradiation conditions, which was done. In Fig. 8 shows that the effect produced by these two drugs is comparable. This allows us to hope that the drug we have developed, enriched with the isotope boron-10, will give a result no worse than borphenylalanine. Then, in these planned experiments, we will need to carry out more detailed measurements, including the dose response and describe this dependence of the linear quadratic model. In these experiments, we did not set ourselves such a task, so we would like not to make any changes to them due to your comment. Hope for your understanding.

Reviewer 2 Report
The manuscript by Popova et al describes an interesting platform based on human serum albumin as a delivery system for boron compounds for BNCT applications. The focus on the labeling of the conjugate is particularly interesting as it allows the in vivo monitoring of the distribution of the construct. Overall the results are complete and well presented.
I suggest accepting the manuscript after minor revisions as follows:
- In the introduction (page 1 line 39, I suggest to state clearly that the selective destruction of tumor cells with preservation of the healthy tissue is strongly dependent on the ratio of boron concentration between tumor and healthy tissues.
- Page 2 line 45 Boronophenylalanine should not be capitalized
- Page 3 line 99: define better what OFI is; also give some information on the limitations of the method, I presume that it can be useful on small animals but there are limitations in depth of fluorescence detection.
- Figure 1: the boron cage containing substituent is defined both in the legend and in the figure as B12 H12, it should be more appropriate to use B12H11 as one of the boron atoms bears a substituent. Do the same throughout the manuscript
- Page 9 lines 289-291: the same information were already stated in the introduction, so I believe they are unnecessary here
- In the supplementary material, I would suggest adding the hard copies of the NMR spectra of the synthesized compounds
Author Response
The authors are grateful to Reviewer #2 for insightful comments on the manuscript. The authors are grateful to Reviewer #2 also for the expressed questions/criticism, it helped to improve our manuscript a lot. We have addressed the issues highlighted by the reviewer and made the appropriate changes in the manuscript in accordance with the reviewer’s comments and suggestions. We highlighted (yellow highlighted text) all changes made when revising the manuscript.
- In the introduction (page 1 line 39, I suggest to state clearly that the selective destruction of tumor cells with preservation of the healthy tissue is strongly dependent on the ratio of boron concentration between tumor and healthy tissues.
Our response, and changes made to the manuscript:
We agree with this comment. The following text “In order for BNCT to become a viable therapeutic option, the radiation dose delivered to the tumor must exceed the background radiation healthy tissue receives from nonspecific neutron absorption [20].” has been added (page 2, lines 70-72):
“In order for BNCT to become a viable therapeutic option, the radiation dose delivered to the tumor must exceed the background radiation healthy tissue receives from nonspecific neutron absorption [20]. As BNCT relies on neutron irradiation of the tumor-accumulated boron compounds for the therapeutic effect to occur, it is important to monitor the drug distribution inside the body in order to determine optimal time window for the irradiation to be performed after drug administration. This can be achieved by introducing a tracking tag into the drug carrier. Examples of such tags include positron or gamma emitters that are incorporated into a nanoparticle-based drug delivering system. In this case, the drug distribution and localization can be monitored using positron emission tomography (PET) or Single Photon Emission Computerised Tomography (SPECT) imaging [41,42]. 18F–BPA positron emission tomography (PET) imaging [43,44] is now a well-established technique used as part of the treatment planning protocols both in Japan and Finland, the two countries where the largest number of patients have been treated by BNCT.”
- Page 2 line 45 Boronophenylalanine should not be capitalized
Our response, and changes made to the manuscript:
We agree with this comment. Corrected (page 2, line 45) ”Boronophenylalanine” was replaced by “boronophenylalanine”.
- Page 3 line 99: define better what OFI is; also give some information on the limitations of the method, I presume that it can be useful on small animals but there are limitations in depth of fluorescence detection.
Our response:
Fluorescence-based molecular imaging becomes increasingly important, mainly due to the development of highly sensitive cameras and very high spatial resolution. We agree with Reviewer comment that it can be useful on small animals but there are limitations in depth of fluorescence detection. Our labelling strategies enable determination of the nanoconjugate location in vivo, in real time and with high sensitivity, thus reducing the number of animals required for fast investigation of new nanosystems as chemotherapeutic agent and BNCT drug candidates.
Moreover, it is often desirable to use the same compound for different clinical applications, e.g., combined pre- and intraoperative tumor detection. Herein, albumin has been functionalized with a combination of a fluorescent dye and fluorinated compound, yielding a dual-labeled molecular probe for 19F MRI and fluorescence microscopy. A potential application for such multimodal compounds could be an integrated use of preoperative diagnostics and surgical planning (magnetic resonance-based), with intraoperative surgical image guidance (fluorescence-based) to the predefined lesion location. This is of acute importance in case of brain tumors. Complete surgical resection of brain tumors is difficult, and methods enabling better discrimination of the tumor borders are needed to maximize therapeutic success while minimizing collateral damage. Fluorescence-guided brain surgery is an accepted new therapeutic option in the treatment of patients with glioma that can result in longer progression-free survival. Indeed, various fluorescent compounds have already been suggested for such (intraoperative) imaging applications, and also as candidates for image-guided therapy, even despite the difficulty in this case due to a limited tissue penetration of fluorescence signals. To link different applications, it is desirable to use one-and-the-same imaging agent, functionalized with multiple diagnostic labels. Simultaneous presence of undecahydro-closo-dodecaborate and labeled amino acid residues (fluorophore dye and fluorine atoms) in the obtained HSA conjugate makes it a promising candidate for the combination imaging‐guided boron neutron capture therapy. However, more detailed studies with albumin conjugate involving toxicity, metabolic fate, and immunologic consequences are necessary to establish its potential as multimodal probe for diagnostic and molecular imaging.
In these experiments, we did not set ourselves such a task, so we would like not to make any changes to text due to your comment. Hope for your understanding.
- Figure 1: the boron cage containing substituent is defined both in the legend and in the figure as B12 H12, it should be more appropriate to use B12H11 as one of the boron atoms bears a substituent. Do the same throughout the manuscript
Our response, and changes made to the manuscript:
We agree that the name of the boron cage containing substituent should be changed. In this connection, in the manuscript the following boron cage B12H12 was replaced to B12H11.
- Page 9 lines 289-291: the same information were already stated in the introduction, so I believe they are unnecessary here
Our response, and changes made to the manuscript:
We agree with this comment. The following text “This compound was previously used as a 10B compound in clinical trials in reactor-based BNCT, along with disodium mercaptoundecahydrododecaborate (BSH) [23].” has been deleted (page 9, lines 289-299):
“Recently, several accelerators destined for hospital placement have been introduced [78]. For BNCT purposes, a proton accelerator with vacuum insulation and a lithium target have been developed at the Budker Institute of Nuclear Physics (BINP) at the Russian Academy of Sciences (Novosibirsk, Russian Federation) [67]. To date, initial radiobiological experiments on tumor cells to evaluate the efficacy of the neutron source at BINP have been performed with L-p-boronophenylalanine (BPA) [79,80], a boron agent. In the current article, we provide in vitro efficacy evaluation of this unique accelerator-based neutron source, for boron neutron capture therapy (BNCT) at BINP, using BPA and our albumin conjugate. We believe that our study will bring small clarity to ongoing in vitro experiments on neutron capture therapy and help other researchers to advance accelerator-based BNCT into the clinical phase.”
- In the supplementary material, I would suggest adding the hard copies of the NMR spectra of the synthesized compounds
Our response, and changes made to the manuscript:
We have added the hard copies of the NMR spectra of the synthesized compounds:
- Maleimide-conjugating closo-dodecaborate tetramethylammonium form (B12H11-mal (Bu4N+));
- 2,2,2-Trifluoro-N-(2-oxotetrahydrothiophen-3-yl)acetamide (HTLTFAc);
- HSA‑Cy5‑HcyAc-B12H11-TTFA;
- HSA‑Cy5‑HcyTFAc-B12H11
in the end of the supplementary material (the chapter “NMR data”).

Reviewer 3 Report
In this work, Popova et al. synthesized the HSA-Cy5-HcyAc-B12H12-TTFA based on the synthesis methodology of albumin conjugates for the combination imaging‐guided boron neutron capture therapy.
The TTFA is both imaging agent and chemotherapeutic agent. Boron is conjugated with HSA which is a carrier protein with a long plasma half-life for extended systemic circulation of the boron compounds and increase their accumulation in human glioma cells.
The conjugate is a new promising albumin-based boron theranostic agent with multimodal functions with each fluorescence imaging (Cy5) or 19F MRI. It is comparable with currently available BNCP like BPA and BSH.
This manuscript is well described and the results are supported with clear data. It is acceptable for publication in Molecules after minor revision with addressing a few comments below.
- Grammar check is required ( ex, will be enable on line 483)
- In Figure 1b, the description for X and Y axis is required.
- In Figure 3, the C on X axis should be addressed.
- In all Figures, unit can be put in the parenthesis. (ex, Cell viability (%))
- The full name should be addressed at the first time abbreviations are used.
- Lower letters should be used in legend of figure 3, figure 7 and figure 8.
- Cell name of U87MG and U-87MG should be unified.
Author Response
The authors are grateful to Reviewer #3 for insightful comments on the manuscript. The authors are grateful to Reviewer #3 also for the expressed questions/criticism, it helped to improve our manuscript a lot. We have addressed the issues highlighted by the reviewer and made the appropriate changes in the manuscript in accordance with the reviewer’s comments and suggestions. We highlighted (yellow highlighted text) all changes made when revising the manuscript.
- Grammar check is required ( ex, will be enable on line 483)
Our response: Corrected.
- In Figure 1b, the description for X and Y axis is required.
Our response, and changes made to the manuscript:
In Figure 1b, the description for X axis was corrected. An X axis has been added designation “δ (ppm)”). The scale Y is not usually required in NMR spectra.
- In Figure 3, the C on X axis should be addressed.
Our response, and changes made to the manuscript:
We have edited the text and Figure 3 in accordance with the conditions of our experiment (page 7, lines 254-255).
Figure 3. Viability of T98G cells. The cancer cell line was incubated for 72 h with HSA (green); HSA‑Cy5-HcyTFAc-B12H11 (cyan); HSA‑Cy5-HcyAc-B12H11-TTFA (gray) at various concentrations. After incubation, the cell viability was measured using the MTT test. The reported values represent the mean ± SD (n = 3). Two-way ANOVA was used for comparisons of more than two sets of data. Differences were considered to be significant if the P-value was <0.05.
- In all Figures, unit can be put in the parenthesis. (ex, Cell viability (%))
Our response: Corrected.
- The full name should be addressed at the first time abbreviations are used.
Our response, and changes made to the manuscript:
The full names are inserted:
- page 1, line 12-13 – “human serum albumin”;
- page 2, line 85 – “computed tomography”;
- page 2, line 86 – “magnetic resonance imaging”.
- Lower letters should be used in legend of figure 3, figure 7 and figure 8.
Our response, and changes made to the manuscript:
We have changed the text to better reflect the findings of our work. The following text “Survival data were fit by a linear regression according to the linear-quadratic formula (LQ fit) [68]. We did not receive any clones after irradiation U87MG cells with pretreatment BPA.” has been added (page 9, lines 304-306):
“The glioma cells were incubated in medium with HSA-Cy5-HcyTFAc-B12H11 conjugate or with BPA and irradiated by epithermal neutron flux. Boron-negative cells, irradiated by neutrons were used as controls. Survival of U87MG tumor cells incubated in medium with HSA-Cy5-HcyTFAc-B12H11 conjugate or with BPA reduced with the increase of neutron flux (Figure 7). Survival data were fit by a linear regression according to the linear-quadratic formula (LQ fit) [68]. We did not receive any clones after irradiation U87MG cells with pretreatment BPA. Even though, the increase of neutron flux affected survival of the cells without boron, pretreatment with HSA-Cy5-HcyTFAc-B12H11 or BPA resulted in the significantly lower fraction of of the survived U87MG cells as compared with the conjugate-free cells (Cntr) (P < 0.0001). Multiple comparisons by two-way ANOVA of the survived fractions did not shown statistical difference between the U87MG cell line pretreated with HSA-Cy5-HcyTFAc-B12H11 and BPA.”
We have edited the text (page 10, lines 313-314 and 325-326) and Figures 7 and 8 in accordance with the conditions of our experiment.
Figure 7. Survival curves of U87MG cells incubated with HSA-Cy5-HcyTFAc-B12H11 conjugate (cyan curve) and without reagents (orange curve) depending on neutron fluence.
Figure 8. The cell viability of U87MG cells incubated with HSA-Cy5-HcyTFAc-B12H11 conjugate and BPA before BNCT at 2, 4, 6 days after neutron irradiation. Control: U87MG cells after neutron irradiation without boron-containing compounds.
- Cell name of U87MG and U-87MG should be unified.
Our response, and changes made to the manuscript:
We agree that the cell name of U87MG and U-87MG should be unified. In this connection, in the manuscript the following cell name of U-87MG was replaced with U87MG.

Reviewer 4 Report
This work makes an important contribution to studies of the application of BNCT for the treatment of untreatable cancers by conventional therapies such the glioma. It is very promising and encouraging for the optimization of BNCT to be able to count on these compounds with so many combined properties. In addition, no less important, is the fact of having carried out the studies using a particle accelerator as a neutron source.
However it would be appropriate to make some corrections for a better interpretation and comparison of results.
1) Regarding Figure 7, no significant differences are observed with respect to boronophenilalanine at any of the evaluated neutron fluences and very low survival fractions. It would be advisable to present the dose response data measured with the linear quadratic model.
In order to be able to compare with data from the literature, it would be advisable to transform the neutron fluences into total physical doses absorbed in Gray.
2) Figure 8 showing survival versus time for the same compounds should be plotted not as optical density but as fraction of survival. In this way it will be easier to interpret the results and compare them with Figure 7.
3) In addition you should discuss how they consider that a borated compound bound to albumin would affect crossing the blood-brain barrier.
4) To demonstrate the selectivity of the new compound, it would be appropriate to carry out uptake studies both in tumor lines and in normal cells. If you have studies carried out, it would be important to comment on them in the discussion or conclusions.
5) It should be noted that these are in vitro studies and that it would be interesting to evaluate the pharmacokinetics of the compound in in vivo biological models. Add a coment about this point.
Author Response
The authors are grateful to Reviewer #4 for insightful comments on the manuscript. The authors are grateful to Reviewer #4 also for the expressed questions/criticism, it helped to improve our manuscript a lot. We have addressed the issues highlighted by the reviewer and made the appropriate changes in the manuscript in accordance with the reviewer’s comments and suggestions. We highlighted (yellow highlighted text) all changes made when revising the manuscript.
- Regarding Figure 7, no significant differences are observed with respect to boronophenilalanine at any of the evaluated neutron fluences and very low survival fractions. It would be advisable to present the dose response data measured with the linear quadratic model. In order to be able to compare with data from the literature, it would be advisable to transform the neutron fluences into total physical doses absorbed in Gray.
- Figure 8 showing survival versus time for the same compounds should be plotted not as optical density but as fraction of survival. In this way it will be easier to interpret the results and compare them with Figure 7.
Our response, and changes made to the manuscript:
We did not set the problem of absolute measurements, therefore we limited ourselves to relative ones, especially since if the dose for BNCT is given, then it will have to be described for all four components: boron, gamma-rays, fast neutrons, nitrogen with RBE, CBE, the numerical values of which are given by different authors are given differently.
We have changed the text to better reflect the findings of our work. The following text “Survival data were fit by a linear regression according to the linear-quadratic formula (LQ fit) [68]. We did not receive any clones after irradiation U87MG cells with pretreatment BPA.” has been added (page 9, lines 304-306):
The glioma cells were incubated in medium with HSA-Cy5-HcyTFAc-B12H11 conjugate or with BPA and irradiated by epithermal neutron flux. Boron-negative cells, irradiated by neutrons were used as controls. Survival of U87MG tumor cells incubated in medium with HSA-Cy5-HcyTFAc-B12H11 conjugate or with BPA reduced with the increase of neutron flux (Figure 7). Survival data were fit by a linear regression according to the linear-quadratic formula (LQ fit) [68]. We did not receive any clones after irradiation U87MG cells with pretreatment BPA. Even though, the increase of neutron flux affected survival of the cells without boron, pretreatment with HSA-Cy5-HcyTFAc-B12H11 or BPA resulted in the significantly lower fraction of of the survived U87MG cells as compared with the conjugate-free cells (Cntr) (P < 0.0001). Multiple comparisons by two-way ANOVA of the survived fractions did not shown statistical difference between the U87MG cell line pretreated with HSA-Cy5-HcyTFAc-B12H11 and BPA.
We have edited the text (page 10, lines 313-314 and 325-326) and Figures 7 and 8 in accordance with the conditions of our experiment.
Figure 7. Survival curves of U87MG cells incubated with HSA-Cy5-HcyTFAc-B12H11 conjugate (cyan curve) and without reagents (orange curve) depending on neutron fluence.
Figure 8. The cell viability of U87MG cells incubated with HSA-Cy5-HcyTFAc-B12H11 conjugate and BPA before BNCT at 2, 4, 6 days after neutron irradiation. Control: U87MG cells after neutron irradiation without boron-containing compounds.
- In addition you should discuss how they consider that a borated compound bound to albumin would affect crossing the blood-brain barrier.
- To demonstrate the selectivity of the new compound, it would be appropriate to carry out uptake studies both in tumor lines and in normal cells. If you have studies carried out, it would be important to comment on them in the discussion or conclusions.
- It should be noted that these are in vitro studies and that it would be interesting to evaluate the pharmacokinetics of the compound in in vivo biological models. Add a comment about this point.
Our response:
Glioma is the most invasive brain tumor with a poor prognosis. There are several critical issues, as itemized below, that must be addressed before applying BNCT as a useful modality for cancers treatment.
- More effective boron-containing agents are required so that they can be used alone or in combination with other agents to deliver the necessary amount of boron to cancer cells.
- The delivery of boron-containing agents to the cancer cells and cellular microdistribution must be optimized to improve its uptake, especially to diverse subpopulation of cancer cells.
- The development of new imaging agents to provide estimates of boron content in the cancer cells are essential. Therefore, the development of new imaging agents is the other fundamental need for the future progress of BNCT.
The goal of our work is to develop a new drug for targeted delivery of boron for BNCT and for provided estimates of boron content in the cancer cells. We demonstrated intracellular uptake of the boron-albumin conjugate by the human glioma cells. The localization of albumin-based boron theranostic agent was confirmed by fluorescence microscopy. In Figs. 5 and 6 show the distribution of HSA conjugate in tumor cells. These results are fully consistent with the design expectations and provide experimental support for the suggestion that homocystamides HSA are attractive candidates for use in boron compounds delivery systems in the human glioma cells. Although the uptake mechanism of albumin by the cells is still not clear except for gp60-mediated transcytosis of endothelial cells (J. E. Schnitzer and P. Oh, 1994) and SPARC facilitated albumin accumulation (N. Desai, et al., 2004), an interesting in vivo study showing evidence of albumin homocystamides uptake has been recently reported (V.A. Lisitskiy, et al., 2017). Lisitskiy et al. found that malignant glioma actively scavenged extracellular proteins, such as albumin conjugates. Our current investigation of localisation of boron-albumin conjugate in the human glioma cells also supports their results. The investigation of the boron-containing albumin conjugate as an in vivo theranostic agent is currently in progress and the results will be reported by us further.
Herein, it was extremely important for us to compare its effect with the effect of the only approved drug, boron-phenylalanine. This drug is not yet final, since we used natural boron with 20% boron-10 content, and not enriched to 98%. At this stage of the research, it was important for us to compare these two drugs under the same irradiation conditions, which was done. In Fig. 8 shows that the effect produced by these two drugs is comparable. This allows us to hope that the drug we have developed, enriched with the isotope boron-10, will give a result no worse than borphenylalanine. Then, in these planned experiments, we will need to carry out more detailed measurements, including the dose response and describe this dependence of the linear quadratic model. In these experiments, we did not set ourselves such a task, so we would like not to make any changes to them due to your comment. Hope for your understanding.

Round 2
Reviewer 1 Report
The revised manuscript is now improved significantly, can be accepted for publication.